# The association between perceived stress with sleep quality, insomnia, anxiety and depression in kidney transplant recipients during Covid-19 pandemic

**Dilek Barutcu Atas[1]\*, Esra Aydin Sunbul[2], Arzu Velioglu[1], Serhan Tuglular[1]**

**1** Division of Nephrology, Department of Internal Medicine, School of Medicine, Marmara University, Istanbul, Turkey, **2** Department of Psychiatry, University of Health Sciences Erenköy Mental Health and Neurological Diseases Training and Research Hospital, Istanbul, Turkey

\* drdilekb@gmail.com

**Data Availability Statement:** All relevant data are within the manuscript and its Supporting Information files.

## Abstract

### Background

The psychological distress and sleep problems caused by current Covid-19 outbreak is not well known in kidney transplant recipients. In this study, we aimed to investigate the association between perceived stress with sleep quality, insomnia, anxiety, depression and kidney function in kidney transplant recipients during the Covid-19 pandemic.

### Material and methods

A hundred-six kidney transplant recipients were enrolled. Questionnaire of "Socio-demographics", "Perceived Stress Scale (PSS)", "Pittsburgh Sleep Quality Index (PSQI)", "Insomnia Severity Index (ISI)" and "Hospital Anxiety Depression Scale (HADS)" are performed. The laboratory data is recorded. The perceived stress related to Covid-19 pandemic and its associations were investigated.

### Results

The mean age of patients was 44.2±13.3 years, and 65 of the patients (61.3%) were men. Forty-nine (46.2%) of the patients had high-perceived stress; 51 (48.1%) of the patients had poor sleep quality, 40 (37.7%) of the patients had insomnia, 25 (23.6%) of the patients had anxiety and 47 (44.3%) of the patients had depression. The patients having a history of Covid-19 infection in own or closed relatives (9.09±4.17 vs 6.49±4.16, p:0.014) and the patients who have a rejection episode any of time (8.24±5.16 vs 6.37±3.57, p:0.033) have had significantly higher anxiety scores, when they compared to others. The high PSS were positively correlated with PSQI, ISI, HAD-A and HAD-D. Regression analyses revealed that high-perceived stress is an independent predictor of anxiety and depression. There was not significant difference between kidney function with PSS, PSQI, ISI, HAD-A and HAD-D.

**Funding:** The author(s) received no specific funding for this work.

**Competing interests:** The authors have declared that no competing interests exist.

## Conclusions

High PSS is positively correlated with poor sleep quality and insomnia and also an independent predictor of anxiety and depression in kidney transplant recipients during the outbreak of Covid-19. As the pandemic is still spreading worldwide quickly early identification and intervention of sleep disturbances and psychiatric disorders are essential to protect graft function with high compliance to treatment in transplantation patients.

## Introduction

Kidney transplantation is the best option for the treatment of end stage renal disease. Kidney transplant recipients are on life long immunosuppressive drugs which render them to be more vulnerable to infectious diseases such as current Covid-19.

The coronavirus disease 2019 (Covid-19), which emerged in China in December 2019, is caused by severe acute respiratory syndrome coronavirus-2 (SARS-CoV-2) [1]. The disease is characterized by a complex highly variable disease pathology mostly including respiratory system and results in significant challenges in organ transplantation. Since March 2020, COVID-19 has spread to all countries worldwide and the World Health Organization declared a pandemic of international concern. As of 30 Dec 2020, there were 80 453 105 confirmed cases and 1 775 776 confirmed deaths worldwide with cases attributable to this disease [2].

In the pandemic period, restrictions in social life, social isolation, quarantine, boredom, inadequate information, and virus-related fears can lead to various psychiatric disorders in susceptible individuals [3].

Studies of previous quarantines for SARS, influenza A and Ebola revealed high rates of post-traumatic stress and depression up to 4 to 5 times higher in quarantined subjects [4]. It has been shown before that transplant recipients are susceptible to anxiety, depression, and post-traumatic stress disorder [5]. Some studies report that post-transplant depression and anxiety increases morbidities and mortality risk of the patients, with poorer medical adherence and/or pathophysiological abnormalities which contribute to poor health outcomes [6,7].

An increase in the risk and severity of infection is expected in kidney transplant recipients due to immunosuppression [8,9]. Besides the fear of death due to increased susceptibility to infections, social distancing measures, difficulties in reaching to hospital/drugs or maintaining the treatment or monitoring the drug level, fear of rejection and returning to dialysis or potential financial losses can act as a further psychosocial stressor in kidney transplant recipients.

Kidney recipients already have to face some challenges before, during and after transplant surgery. According to the American Psychological Association, anxiety or depression episode are prevalent in 50% of transplant recipients within two years after transplantation [10]. Such psychiatric disorders may lead to noncompliance to treatment, resulting in graft loss [11]. Non-adherent patients were seven times more at risk of graft failure than adherent patients [12]. Sleep complaints are also common among patients who have received kidney transplantation [13]. Recently, an increased prevalence of sleep disorders and anxiety in health care professionals and general population have been reported during the Covid-19 lockdown [14,15]. However, there isn't any study addressing the latter issue in kidney recipient populations. Outbreaks of infectious diseases and current Covid-19 may trigger significant sleep problems and major psychiatric problems including anxiety, depression [16]. Emotional well-being is important in kidney transplant recipients and improve the treatment compliance and decrease the probability of rejection [17]. Therefore, we aimed to investigate the association between

perceived stresses with sleep quality, insomnia, anxiety, depression and kidney function in kidney transplant patients during the Covid-19 pandemic.

## Methods and materials

### Study design

A hundred-six kidney transplant patients were evaluated in this cross-sectional study. Patients with a functioning kidney graft of at least three months' duration, over 18 years old, able to respond to questionnaires, and no psychiatric disease were included in the study. The study population was selected among kidney transplant patients followed up at the Marmara University Hospital Transplantation clinic between 01.09.2020 and 01.12.2020. The questionnaire forms were distributed to the patients during the scheduled clinic visits, and they filled them out themselves. Participants filled out demographic information on age, gender, marital status, education status, socioeconomic status, working status, smoking, alcohol consumption, causes and duration of chronic kidney disease, duration of kidney transplantation, any history of rejection attack, history of Covid-19 infection and any problem in reaching to the hospital during the pandemic. The laboratory data including glucose, blood urea nitrogen (BUN), creatinine, serum albumin, alanine transaminase, aspartate transaminase, sodium, potassium, serum calcium, phosphorus, complete blood count and proteinuria were recorded at the time of the outpatient clinic follow-up as a part of routine care. The association between perceived stress with sleep quality, insomnia, anxiety, depression and kidney function in kidney transplant patients during the Covid-19 pandemic was analyzed.

### Study survey

Questionnaires of "Socio-demographics," "The Perceived Stress Scale", "Pittsburgh Sleep Quality Index", "Insomnia Severity Index" and "Hospital Anxiety Depression Scale" were performed.

**Perceived stress scale.**   Cohen et al. developed the PSS in 1983 [18]. Consisting of 14 items, the PSS was designed to measure how some stressful situations in a person's life are perceived. PSS indicates stress as perceived during the last month. Participants evaluate each item on a 5-point Likert-type scale ranging from "Never (0)" to "Very often (4)". 7 items that contain positive statements are scored in reverse. The scores of the PSS-14 vary between 0 and 56, and the high score indicates a high perception of stress (cut-off point >25). Reliability and validity were analyzed by adapting the PSS form to Turkish [19].

**Pittsburgh sleep quality index.**   The Pittsburgh Sleep Quality Index (PSQI) is a self-rated questionnaire that assesses sleep quality and disturbances over a 1-month time interval. Nineteen individual items generate seven "component" scores: subjective sleep quality, sleep latency, sleep duration, habitual sleep efficiency, sleep disturbances, use of sleeping medication, and daytime dysfunction. A total score, ranging from 0 to 21, is obtained by adding the 7 component scores. A global PSQI score greater than 5 indicates a "poor" sleep quality. The clinometric and clinical properties of the PSQI suggest its utility both in psychiatric clinical practice and research activities [20]. The validity and consistency (test-retest reliability) of the PSQI was adopted to Turkish by Ağargün et al, in 1996 [21].

**Insomnia severity index.**   The ISI is a brief instrument that was designed to assess the severity of both night-time and day-time components of insomnia. ISI was developed by Morin and has been translated into various languages [22]. Boysan et al. evaluated psychometric properties of the Insomnia Severity Index in a Turkish sample, and they found that the ISI revealed adequate validity and reliability [23]. The ISI is a 7-item self-report questionnaire assessing the nature, severity, and impact of insomnia [24]. The usual recall period is the "last

month" and the dimensions evaluated are: severity of sleep onset, sleep maintenance, and early morning awakening problems, sleep dissatisfaction, interference of sleep difficulties with daytime functioning, noticeability of sleep problems by others, and distress caused by the sleep difficulties. A 5-point Likert scale is used to rate each item (e.g., 0 = no problem; 4 = very severe problem), yielding a total score ranging from 0 to 28. The total score is interpreted as follows: the absence of insomnia (0–7); sub-threshold insomnia (8–14); moderate insomnia (15–21); and severe insomnia (22–28) [25].

**The hospital anxiety and depression scale.** This scale is a widely used self-rating scale for determination of patient's depression and anxiety status [26]. HADS consists of 14 items, and 7 items for depression (HAD-D) and seven items for anxiety (HAD-A). All items with a four-point ordinal scale to describe symptom severity: from zero (not present) to three points (strongly present). HADS-D mainly covers the core depressive symptoms of anhedonia and loss of interest. HADS-A mainly covers the core anxiety features worry and tension. Both subscales by design exclude somatic components of depression and anxiety. The reliability and validity of the Turkish version of HADS have been established in Turkish patients, and it has been reported that the cut-off point in the anxiety subscale score is 10 and in the depression subscale score is 7 [27].

The investigation conforms with the principles outlined in the Declaration of Helsinki. The study design was approved by the institutional review board of Marmara University School of Medicine Ethic Committee and all participants gave written informed consent. (Protocol number: 09.2020.991).

## Statistical analysis

SPSS (version 22.0; SPSS Inc, Chicago, IL) statistics package was used for statistical analysis. Categorical variables were presented as numbers and percentages and compared with the Chi-square test. Continuous variables were presented as mean ± standard deviation. Continuous variables with parametric distribution were compared with independent samples t-test, and those without normal distribution were compared with Mann-Whitney U-test. Kolmogorov-Smirnov analysis was performed to determine whether continuous variables were normally distributed. According to the normality tests, those with $p \geq 0.05$ were considered to be normally distributed. The Pearson or Spearman correlation test was used where appropriate. Logistic regression analyses were performed to determine independent predictors of high perceived stress in kidney transplantation patients. For all statistical analyses, a p-value $<0.05$ was considered significant.

## Results

The mean age of patients was 44.2±13.3 years, and 65 of the patients (61.3%) were male. Most of the patients (91.5%) had first transplantation. The mean duration of kidney transplantation was 7.7±5.6 years. Thirty-eight (35.8%) of the patients had a history of rejection and only 5 (4.7%) of the patients had a history of Covid-19 infection and all of them improved. Thirty (28.3%) of the patients had difficulty in reaching to the hospital during the Covid-19 outbreak. The baseline characteristics and clinical data of patients are summarized in Table 1.

Data presented as mean ± standard deviation.

The mean PSS of the study population was 25.1±8.1; the mean PSQI score was 6.0±5.5; mean ISI score was 6.6±5.5. The mean HADS score was 7.1±4.3 for anxiety and 6.4±3.6 for depression. According to the scales, 49 (46.2%) of the patients had high perceived stress; 51 (48.1%) of the patients had poor sleep quality, 40 (37.7%) of the patients had insomnia, 25 (23.6%) of the patients had anxiety and 47 (44.3%) of the patients had depression.

**Table 1. Baseline characteristics and clinical data of study population (n = 106).**

| | |
|---|---|
| **Age (years)** | 44.2±13.3 |
| **Sex—male–(n-%)** | 65 (61.3%) |
| **Smoking (n-%)** | 7 (6.6%) |
| **Alcohol (n-%)** | 7 (6.6%) |
| **Body mass index (kg/m²)** | 25.5±4.5 |
| **Duration of kidney transplantation (years)** | 7.7±5.6 |
| **Duration of Chronic Kidney Disease (years)** | 6.6±6.9 |
| **Marital Status—married (n-%)** | 84 (79.2%) |
| **Education Status—literate (n-%)** | 103 (97.2%) |
| **Economic Status—Low *(n-%)*** | 16 (15.1%) |
| **Working (n-%)** | 29 (72.6%) |
| **Living with (n-%) *Alone*** | 8 (7.5%) |
| *Nuclear family* | 78 (73.6%) |
| *Extended family* | 20 (18.9%) |
| **First Transplantation (n-%)** | 97 (91.5%) |
| **History of rejection (n-%)** | 38 (35.8%) |
| **History of Covid-19 (n-%)** | 5 (4.7%) |
| **Covid-19 History of Family Member (n-%)** | 19 (17.9%) |
| **Difficulty to reaching the hospital (n-%)** | 30 (28.3%) |
| **Presence of high perceived stress (n-%)** | 49 (46.2%) |
| **Presence of poor sleep quality (n-%)** | 51(48.1%) |
| **Presence of insomnia (n-%) *Sub-threshold*** | 33 (31.1%) |
| *Moderate* | 5 (4.7%) |
| *Severe* | 2 (1.9%) |
| **Presence of anxiety (n-%)** | 25 (23.6%) |
| **Presence of depression (n-%)** | 47 (44.3%) |
| **Perceived Stress Scale score** | 25.1±8.1 |
| **Total Pittsburgh Sleep Quality Index score** | 6.0±5.5 |
| **Insomnia Severity Index score** | 6.6±5.5 |
| **Hospital Anxiety Depression Scale-Anxiety score** | 7.1±4.3 |
| **Hospital Anxiety Depression Scale-Depression score** | 6.4±3.6 |

Patients with PSS score ≥25 was accepted as high PSS score group. While 49 (46.2%) patients had a high PSS score, 57 (53.8%) patients had a low PSS score. The total PSQI score was significantly higher in high PSS score than low PSS score (7.1±4.3 vs 5.0±3.1, p: 0.006). Higher ISI scores were significantly related to high PSS score than low PSS score (8.4±6.5 vs 4.9±4.0, p:0.003) during the Covid-19 pandemic. HADS-A score (9.0±4.2 vs 5.2±3.6, p: <0.001) and HADS-D score (7.8±3.1 vs 5.2±3.5, p: <0.001) were significantly higher in patients with high PSS score compared to low PSS score. Age and sex were similar between groups. Most of the patients (77.6%) in the high PSS score group were members of a nuclear family. Comparison of baseline characteristics and clinical data according to PSS score is shown in Table 2.

The mean serum creatinine level of the study population was 1.6±0.9 and 44 (41.5%) of the patients had proteinuria. The laboratory findings of the patients according to PSS were summarized in Table 3.

When patients with less than 60 ml/min/1.73 m² glomerular filtration rate (GFR) were compared to those with GFR>60 ml/min/1.73 m²; there were no significant differences in PSS (24.8±8.2 vs 24.5±9.4, p: 0.877), PSQI (6.2±3.5 vs 5.7±4.0, p: 0.547), ISI (7.3±5.4 vs 5.7±5.7, p:

**Table 2. Comparison of baseline characteristics and clinical data according to Perceived Stress Scale (PSS) score.**

| | Low PSS (n = 57) | High PSS (n = 49) | P |
|---|---|---|---|
| Age (years) | 43.9±13.2 | 44.5±13.5 | 0.664 |
| Sex—male–(n-%) | 39 (68.4%) | 26 (53.1%) | 0.105 |
| Duration of kidney transplantation (years) | 7.5±5.3 | 8.1±6.0 | 0.710 |
| Marital Status—married (n-%) | 45 (78.9%) | 39 (79.6%) | 0.935 |
| Education Status—literate (n-%) | 56 (98.2%) | 47 (95.9%) | 0.595 |
| Economic Status—Low (n-%) | 7 (12.3%) | 9 (18.4%) | 0.383 |
| Working (n-%) | 17 (29.8%) | 12 (24.5%) | 0.539 |
| Living with (n-%) | | | |
| Alone | 6 (10.5%) | 2 (4.1%) | |
| Nuclear family | 40 (70.2%) | 38 (77.6%) | 0.437 |
| Extended family | 11 (19.3%) | 9 (18.4%) | |
| History of rejection (n-%) | 22 (38.6%) | 16 (32.7%) | 0.525 |
| History of Covid-19 (n-%) | 2 (3.5%) | 3 (6.1%) | 0.660 |
| Covid-19 History of Family Member (n-%) | 8 (14.0%) | 11 (22.4%) | 0.260 |
| Difficulty to reaching the hospital | 12 (21.1%) | 18 (36.7%) | 0.074 |
| Total Pittsburgh Sleep Quality Index Score | 5.0±3.1 | 7.1±4.3 | **0.006** |
| Insomnia severity Index Score | 4.9±4.0 | 8.4±6.5 | **0.003** |
| Hospital Anxiety Depression Scale-Anxiety | 5.2±3.6 | 9.0±4.2 | **<0.001** |
| Hospital Anxiety Depression Scale-Depression | 5.2±3.5 | 7.8±3.1 | **<0.001** |

PSS: Perceived stress scale. Data presented as mean ± standard deviation.

0.168), HAD-A (7.4±4.6 vs 6.5±4.1, p: 0.357) and HAD-D (6.4±3.7 vs 6.7±3.6, p: 0.682) scores between the two groups. There were no correlations between glomerular filtration rate and PSS (r: -0.047, p: 0.646), PSQI (r: -0.037, p: 0.725), ISI (r: -0.164, p: 0.115), HAD-A (r: -0.085, p: 0.414) and HAD-D (r: 0.021, p: 0.843).

When patients with hypertension (n:17) were compared to those without hypertension (n:89); there were no significant differences in PSS (24.3±6.4 vs 24.9±8.6, p: 0.784), PSQI (6.6 ±4.0 vs 6.0±3.8, p: 0.547), ISI (7.3±5.0 vs 6.5±5.7, p: 0.597), HAD-A (8.0±4.9 vs 7.0±4.1, p: 0.500) and HAD-D (7.8±3.2 vs 6.2±3.6, p: 0.114) scores between the two groups.

**Table 3. Laboratory parameters of the study population according to Perceived Stress Scale (PSS) score.**

| | Low PSS (n:57) | High PSS (n:49) | p |
|---|---|---|---|
| Hemoglobin, g/dL | 12.9±2.1 | 12.8±2.2 | 0.964 |
| Glucose, mg/dL | 101.2±33.2 | 106.1±52.1 | 0.590 |
| Blood urea nitrogen, mg/dL | 26.3±16.6 | 24.8±12.6 | 0.631 |
| Creatinine, mg/dL | 1.67±1.07 | 1.45±0.49 | 0.222 |
| GFR, ml/min/1.73 m$^2$ | 58.6±28.4 | 55.7±20.4 | 0.565 |
| Albumin, g/dL | 4.2±0.5 | 4.1±0.5 | 0.788 |
| Sodium, mEq/L | 139.6±3.0 | 139.7±3.2 | 0.809 |
| Potassium, mEq/L | 4.4±0.6 | 4.5±0.6 | 0.211 |
| Calcium, mg/dL | 9.5±0.8 | 9.6±0.7 | 0.508 |
| Phosphorus, mg/dL | 3.5±1.0 | 3.4±0.9 | 0.693 |
| Proteinuria (n-%) | 22 (38.6%) | 22 (44.8%) | 0.135 |

PSS: Perceived stress scale. Data presented as mean ± standard deviation.

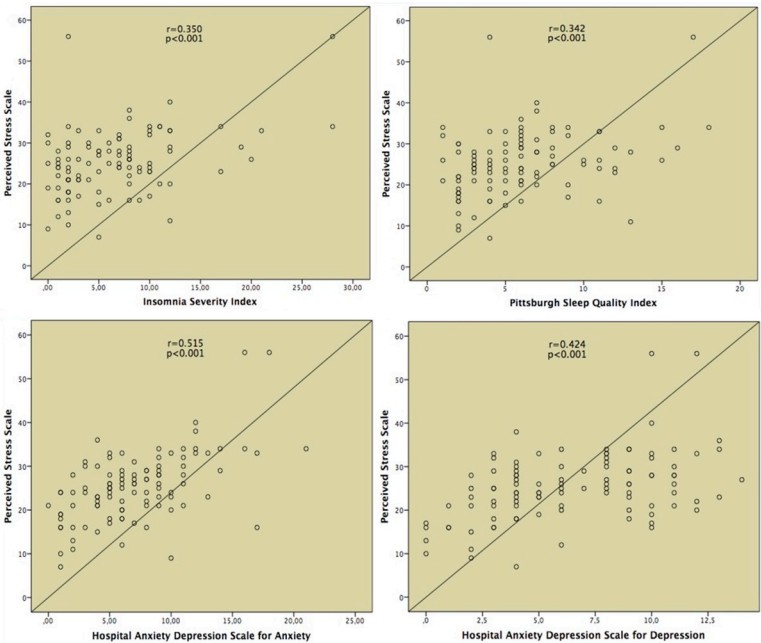

**Fig 1. Correlation analysis between perceived stress with PSQI, ISI, HAD-A and HAD-D score.**

The patients having a history of Covid-19 infection in own or closed relatives (9.09±4.17 vs 6.49±4.16, p: 0.014) and the patients who had a rejection episode at any time (8.24±5.16 vs 6.37±3.57, p: 0.033) have had significantly higher HAD-A scores, when they compared to others, respectively.

The correlation analysis was performed to reveal the association between perceived stress with PSQI, ISI and HADS score. The high PSS scores were positively correlated with PSQI, ISI, HAD-A and HAD-D score (Fig 1).

Multivariate logistic regression analyses revealed that high perceived stress is associated with HAD-A and HAD-D score (Table 4).

## Discussion

To our knowledge this is the first study that investigated Covid-19 associated perceived stress with sleep disturbances and psychiatric disorders in kidney transplant recipients. The findings of the present study offer important implications for the psychiatric management of renal transplant recipients. The results indicate that; kidney transplant recipients have a high level of Covid-19 associated perceived stress, they experience insomnia, they have poor sleep quality, and they have more anxiety and depression, during the Covid-19 pandemic. Patients who

**Table 4. Multivariate logistic regression analysis to determinate predictors of high perceived stress scale score.**

|  | Odds Ratio | 95% Confidence Interval | p |
|---|---|---|---|
| **ISI Score** | 0.979 | 0.830–1.155 | 0.802 |
| **Total PSQI Score** | 1.072 | 0.872–1.319 | 0.509 |
| **HADS-Anxiety score** | 1.212 | 1.047–1.403 | **0.010** |
| **HADS-Depression score** | 1.173 | 1.014–1.357 | **0.032** |

**PSS:** Perceived stress; **HADS:** Hospital Anxiety Depression Scale; **ISI:** Insomnia severity index; **PSQI:** Pittsburgh Sleep Quality Index.

experienced Covid-19 themselves or with closed relatives and a rejection episode at any time have more perceived stress associated anxiety.

The outbreak of Covid-19 is a new and highly evolving stressor for everyone including people with chronic conditions as well as kidney transplant recipients due to disruptions in daily life, social interactions and negative emotions. Huang et al have demonstrated that the prevalence of depression and anxiety of the public were 20.1% and 35.1% respectively during Covid-19 outbreak [15]. Umucu et al implemented a survey of 269 individuals with chronic conditions to describe the perceived stress levels and coping mechanisms related to COVID-19 and they have reported that participants with chronic conditions have a moderate level of stress, depression, and anxiety [28]. In a recent study in Italian population it was demonstrated that 24.2% of patients with chronic conditions had depression and 19.6% of the patients had anxiety during pandemic [29]. In another study from Italy Diamanti et al investigated psychological distress in 100 patients with autoimmune arthritis and 100 controls during the Covid-19 pandemic. They found that the percentages of increased stress scores (46% vs 32%), depression (42% vs 36%) and anxiety (38% vs 25%) of arthritis patients were significantly higher than controls [30]. In our study group, 46.2% of kidney transplant recipients had high-perceived stress related to Covid-19, 44.3% of the patients had depression, 23.6% of the patients had anxiety.

Although kidney transplantation is the best therapeutic approach for end-stage renal disease, most transplant recipients are coping with stressful factors due to increased susceptibility to infectious disease, frequent drug level monitoring, repeated blood tests and fear of rejection. Another potential explanation for the high perceived stress in kidney recipients may be having a greater risk for developing more severe complications from COVID-19 [31]. The effective treatment of Covid-19 is still controversial. Nacif et al performed a systematic review of solid organ transplantation patients infected with Covid-19 and they reported the mortality rate was 17.4% in kidney transplant recipients [32]. Studies have revealed that greater levels of perceived stress are associated with poorer health status, quality of life, and higher levels of depression, anxiety and functional limitations in individuals with disabilities [33].

The Covid-19 pandemic has had detrimental impacts on physical, mental and social health in the general population, with fears of infection, frustration, boredom, inadequate information, financial loss and stigma identified as stressors [4]. Sleep complaints are common in patients with end stage renal disease [34]. Although the rates of sleep disorders including insomnia tend to decrease after kidney transplantation, it remains elevated compared to the general population [13]. Therefore, transplanted patients tend to be more affected from stress disorders, insomnia, and poor sleep quality even before the Covid-19 pandemic. Gualano et al showed that 46.2% of Italian population with chronic conditions had insomnia during the Covid-19 outbreak [29]. Huang et al have demonstrated that the prevalence of poor sleep quality of the public was 18.2%, during Covid-19 outbreak [15]. In our cohort; 37.7% of the patients have insomnia, 48.1% of the patients have poor sleep quality during the Covid-19 outbreak.

According to our findings, it is not surprising that patients with a history of Covid-19 infection themselves or in close relatives and patients who had a rejection episode at any time had significantly higher anxiety scores.

Previous studies suggest that psychological distress and perceived stress affect compliance negatively and non-compliance results in increased morbidity and mortality in kidney transplant recipients [6,7,11]. Although an investigation about sleep quality and its related psychosocial variables among 438 renal transplant patients showed that the global PSQI scores were higher in participants with abnormal renal function compared with participants with normal renal function [35], there was no correlation between kidney function and PSS, PSQI, ISI, HAD-A and HAD-D scores in our study. Re-evaluation in short and long term follow-up of

our study subjects during and after the Covid-19 pandemic will better reflect the association between sleep quality, stress levels and kidney transplant function.

To address the impact of hypertension due to increased stress on kidney function we analysed the possible association between stress levels, sleep disturbances and hypertension. However, we failed to show any significant association between stress levels, sleep disturbances and hypertension. Xie et al. showed that the global PSQI scores were higher in the hypertension group compared with those in the non-hypertension group [35], The reason for the lack of association in our study may be due to the scarcity of hypertension among our subjects. Future studies may be needed to show the association after Covid-19 pandemic is over."

The major limitation of our study was the small sample size and its cross-sectional nature. All data was collected at one-time point. Since the patients' psychological status and sleep conditions before the out-break were not evaluated, it is difficult to infer a causal relationship between the variables of interest and the latter. Moreover, we could not distinguish the effect of non-Covid-19 factors on perceived high stress, poor sleep quality, insomnia, anxiety and depression. Regarding the methodological limitations, it is important to note that results presented here are preliminary and need to be interpreted with caution. We think our study results would help to understand the perceived stress and its associations during the Covid-19 pandemic. It also highlights how kidney transplantation patients may be at risk of increased perceptions of stress related to Covid-19 as well as sleep disorders and psychological distress.

## Conclusions

The current study demonstrated that high perceived stress is positively correlated with poor sleep quality and insomnia and also an independent predictor of anxiety and depression in kidney transplant patients during the outbreak of Covid-19. As the Covid-19 pandemic is still spreading worldwide early identification and intervention of sleep disturbances and psychiatric disorders are essential to protect graft function with high compliance to treatment in transplantation patients during the Covid-19 pandemic. Therefore, specific strategies should be adopted to cope with perceived stress by closely working with psychiatry team.

## Supporting information

**S1 File.**
(DOCX)

**S2 File.**
(PDF)

**S3 File.**
(PDF)

**S4 File.**
(PDF)

**S5 File.**
(PDF)

## Author Contributions

**Conceptualization:** Dilek Barutcu Atas, Esra Aydin Sunbul, Arzu Velioglu, Serhan Tuglular.

**Data curation:** Esra Aydin Sunbul.

**Formal analysis:** Dilek Barutcu Atas, Esra Aydin Sunbul, Arzu Velioglu, Serhan Tuglular.

**Investigation:** Arzu Velioglu, Serhan Tuglular.

**Methodology:** Dilek Barutcu Atas, Esra Aydin Sunbul, Arzu Velioglu, Serhan Tuglular.

**Resources:** Dilek Barutcu Atas, Arzu Velioglu, Serhan Tuglular.

**Supervision:** Dilek Barutcu Atas, Esra Aydin Sunbul, Arzu Velioglu, Serhan Tuglular.

**Validation:** Dilek Barutcu Atas, Esra Aydin Sunbul, Arzu Velioglu, Serhan Tuglular.

**Writing – original draft:** Dilek Barutcu Atas, Arzu Velioglu, Serhan Tuglular.

**Writing – review & editing:** Dilek Barutcu Atas, Esra Aydin Sunbul, Arzu Velioglu, Serhan Tuglular.

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
