## [Decision Letter · Decision Letter 0]

29 Jan 2021

PONE-D-20-40967

The Association Between Perceived Stress Related with Sleep Quality, Insomnia, Anxiety and Depressionin Kidney Transplant Recipients During the Covid-19 Pandemic

PLOS ONE

Dear Dr. Atas,

Thank you for submitting your manuscript to PLOS ONE. After careful consideration, we feel that it has merit but does not fully meet PLOS ONE’s publication criteria as it currently stands. Therefore, we invite you to submit a revised version of the manuscript that addresses the points raised during the review process.

The authors need to address all Reviewers' comments: Reviewer # 1: This paper was aimed at providing an insight into the impact of the current pandemic on psychological well-being and associated sleep quality in kidney transplant recipients. Sleep quality and stress levels were measured by having patients fill out corresponding questionnaires during clinical visits, and the responses were analyzed in combination with laboratory test results. Overall, the paper addresses a novel and interesting question. As is, the paper's message is that high perceived stress is associated with other psychological indicators such as bad sleep and high anxiety. However, without showing that the psychological impact of the pandemic is associated with increased graft loss risk or any other significant outcome in patients, or at least that the increased stress levels contribute to health issues in patients, it has moderate significance. My main concern is the authors need to more clearly show the connection between sleep quality, stress levels and kidney transplant function, in order to make a case for significance of this study. The importance of the study can at least be shown by citing previous research showing the importance of emotional well-being in kidney transplant recipients overall. Though a statement about this connection is made in abstract and introduction, there were no references provided. In addition, whether the stress levels are impacted by the Covid-19 pandemic was also shown, with moderate to weak evidence. Finally, lack of correlation between PSS/PSQL/ISI/HAD readouts and kidney function from laboratory tests, shown in the results, does not add to the importance of the paper. Going into more detailed analysis, it would be very interesting to see any possible association between stress levels and hypertension, whose impact on kidney function is widely recognized. And to make the paper more useful, it would help to identify what exactly about the pandemic (whether it is lack of human interaction, fear of socializing, loss of job, loss in the family) makes kidney transplant recipients prone to psychological disorders. My smaller notes would be about methods description and results presentation. It would be nice to know at which transplant center(s) the patients included in the study were followed-up. It would be also nice to see the breakdown of stress/sleep scores among patients grouped by creatinine/BUN levels. Most of tables should better be re-designed to focus on differences that are significant, rather than showing everything. Table 3 can be presented such that patients are grouped by GFR and/or by measured stress levels. Some statements (such as “psychiatric disorders may lead to noncompliance” and “Outbreaks of infectious diseases and current Covid-19 may trigger…” in introduction) need to be provided with references. Reviewer # 2: Is a very interesting paper looking at sleep disturbances and psychological issues in renal transplant recipients during the early phases of Covid 19 The authors study is subset of renal transplant recipients early in 2020. This was early in the COVID-19 pandemic. It is unclear what the baseline rate of sleep disturbances and concerns would be in this set of patients. For instance "post holiday blues", other concerns with the renal transplant as well as family concerns

We look forward to receiving your revised manuscript.

Kind regards,

Stanislaw Stepkowski

Academic Editor

PLOS ONE

Journal Requirements:

2. Thank you for your ethics statement: "The investigation conforms with the principles outlined in the Declaration of Helsinki. The local ethics committee approved the study, and all participants gave written informed consent (Protocol number: 09.2020.991)."

Additional Editor Comments:

The authors need to address all Reviewers' comments:

Reviewer # 1:

This paper was aimed at providing an insight into the impact of the current pandemic on psychological well-being and associated sleep quality in kidney transplant recipients. Sleep quality and stress levels were measured by having patients fill out corresponding questionnaires during clinical visits, and the responses were analyzed in combination with laboratory test results. Overall, the paper addresses a novel and interesting question. As is, the paper's message is that high perceived stress is associated with other psychological indicators such as bad sleep and high anxiety. However, without showing that the psychological impact of the pandemic is associated with increased graft loss risk or any other significant outcome in patients, or at least that the increased stress levels contribute to health issues in patients, it has moderate significance.

My main concern is the authors need to more clearly show the connection between sleep quality, stress levels and kidney transplant function, in order to make a case for significance of this study. The importance of the study can at least be shown by citing previous research showing the importance of emotional well-being in kidney transplant recipients overall. Though a statement about this connection is made in abstract and introduction, there were no references provided. In addition, whether the stress levels are impacted by the Covid-19 pandemic was also shown, with moderate to weak evidence. Finally, lack of correlation between PSS/PSQL/ISI/HAD readouts and kidney function from laboratory tests, shown in the results, does not add to the importance of the paper. Going into more detailed analysis, it would be very interesting to see any possible association between stress levels and hypertension, whose impact on kidney function is widely recognized. And to make the paper more useful, it would help to identify what exactly about the pandemic (whether it is lack of human interaction, fear of socializing, loss of job, loss in the family) makes kidney transplant recipients prone to psychological disorders.

My smaller notes would be about methods description and results presentation. It would be nice to know at which transplant center(s) the patients included in the study were followed-up. It would be also nice to see the breakdown of stress/sleep scores among patients grouped by creatinine/BUN levels. Most of tables should better be re-designed to focus on differences that are significant, rather than showing everything. Table 3 can be presented such that patients are grouped by GFR and/or by measured stress levels. Some statements (such as “psychiatric disorders may lead to noncompliance” and “Outbreaks of infectious diseases and current Covid-19 may trigger…” in introduction) need to be provided with references.

Reviewer # 2:

Is a very interesting paper looking at sleep disturbances and psychological issues in renal transplant recipients during the early phases of Covid 19

The authors study is subset of renal transplant recipients early in 2020.

This was early in the COVID-19 pandemic.

It is unclear what the baseline rate of sleep disturbances and concerns would be in this set of patients. For instance "post holiday blues", other concerns with the renal transplant as well as family concerns

Reviewers' comments:

Reviewer's Responses to Questions

**Comments to the Author**

1. Is the manuscript technically sound, and do the data support the conclusions?

Reviewer #1: Partly

Reviewer #2: No

2. Has the statistical analysis been performed appropriately and rigorously? 

Reviewer #1: I Don't Know

Reviewer #2: Yes

3. Have the authors made all data underlying the findings in their manuscript fully available?

Reviewer #1: Yes

Reviewer #2: Yes

4. Is the manuscript presented in an intelligible fashion and written in standard English?

Reviewer #1: Yes

Reviewer #2: No

5. Review Comments to the Author

Reviewer #1: Is a very interesting paper looking at sleep disturbances and psychological issues in renal transplant recipients during the early phases of Covid 19

The authors study is subset of renal transplant recipients early in 2020.

This was early in the COVID-19 pandemic.

It is unclear what the baseline rate of sleep disturbances and concerns would be in this set of patients. For instance "post holiday blues", other concerns with the renal transplant as well as family concerns etc.

Reviewer #2: This paper was aimed at providing an insight into the impact of the current pandemic on psychological well-being and associated sleep quality in kidney transplant recipients. Sleep quality and stress levels were measured by having patients fill out corresponding questionnaires during clinical visits, and the responses were analyzed in combination with laboratory test results. Overall, the paper addresses a novel and interesting question. As is, the paper's message is that high perceived stress is associated with other psychological indicators such as bad sleep and high anxiety. However, without showing that the psychological impact of the pandemic is associated with increased graft loss risk or any other significant outcome in patients, or at least that the increased stress levels contribute to health issues in patients, it has moderate significance.

My main concern is the authors need to more clearly show the connection between sleep quality, stress levels and kidney transplant function, in order to make a case for significance of this study. The importance of the study can at least be shown by citing previous research showing the importance of emotional well-being in kidney transplant recipients overall. Though a statement about this connection is made in abstract and introduction, there were no references provided. In addition, whether the stress levels are impacted by the Covid-19 pandemic was also shown, with moderate to weak evidence. Finally, lack of correlation between PSS/PSQL/ISI/HAD readouts and kidney function from laboratory tests, shown in the results, does not add to the importance of the paper. Going into more detailed analysis, it would be very interesting to see any possible association between stress levels and hypertension, whose impact on kidney function is widely recognized. And to make the paper more useful, it would help to identify what exactly about the pandemic (whether it is lack of human interaction, fear of socializing, loss of job, loss in the family) makes kidney transplant recipients prone to psychological disorders.

My smaller notes would be about methods description and results presentation. It would be nice to know at which transplant center(s) the patients included in the study were followed-up. It would be also nice to see the breakdown of stress/sleep scores among patients grouped by creatinine/BUN levels. Most of tables should better be re-designed to focus on differences that are significant, rather than showing everything. Table 3 can be presented such that patients are grouped by GFR and/or by measured stress levels. Some statements (such as “psychiatric disorders may lead to noncompliance” and “Outbreaks of infectious diseases and current Covid-19 may trigger…” in introduction) need to be provided with references.

6. PLOS authors have the option to publish the peer review history of their article (what does this mean?). If published, this will include your full peer review and any attached files.

Reviewer #1: **Yes: **Deepak Malhotra

Reviewer #2: **Yes: **Dulat Bekbolsynov

---

## [Author Response · Author response to Decision Letter 0]

12 Feb 2021

12.02.2021

Dear Editor in Chief

In response to your letter dated 29.01.2021, we have revised our manuscript entitled “The Association Between Perceived Stress Related with Sleep Quality, Insomnia, Anxiety and Depression in Kidney Transplant Recipients During the Covid-19 Pandemic” taking into account the reviewer’s comments and proposals. Please find below our itemized response to the points raised by the reviewer. 

 Please inform us if you require further revision. Thank you for your kind interest. 

Yours sincerely,

Dilek Barutcu Atas, MD

Explanations to the Comments and Suggestions Raised by the Reviewer 

Reviewer1: 

This paper was aimed at providing an insight into the impact of the current pandemic on psychological well-being and associated sleep quality in kidney transplant recipients. Sleep quality and stress levels were measured by having patients fill out corresponding questionnaires during clinical visits, and the responses were analysed in combination with laboratory test results. Overall, the paper addresses a novel and interesting question. As is, the paper's message is that high perceived stress is associated with other psychological indicators such as bad sleep and high anxiety. 

1. However, without showing that the psychological impact of the pandemic is associated with increased graft loss risk or any other significant outcome in patients, or at least that the increased stress levels contribute to health issues in patients, it has moderate significance.

Response: We would like to thank the reviewer for the kind comments related to our manuscript. We added a paragraph into the introduction section Line 75-81 as follows “Studies of previous quarantines for SARS, influenza A and Ebola revealed high rates of post-traumatic stress and depression up to 4 to 5 times higher in quarantined subjects (4). It has been shown before that transplant recipients are susceptible to anxiety, depression, and post-traumatic stress disorder (5). Some studies report that post-transplant depression and anxiety increases morbidities and mortality risk of the patients, with poorer medical adherence and/or pathophysiological abnormalities which contribute to poor health outcomes (6,7).” And we added a sentence into the introduction section Line 90-91 as follows “Non-adherent patients were seven times more at risk of graft failure than adherent patients (12).

2. My main concern is the authors need to more clearly show the connection between sleep quality, stress levels and kidney transplant function, in order to make a case for significance of this study. 

Response: We would like to thank reviewer for the kind comments related with our manuscript. We completely agree with the reviewer accordingly we added to the Results section line 283-285 as follows “There were no correlations between glomerular filtration rate and PSS (r: -0.047, p: 0.646), PSQI (r: -0.037, p: 0.725), ISI (r: -0.164, p: 0.115), HAD-A (r: -0.085, p: 0.414) and HAD-D (r: 0.021, p: 0.843).” And we added to the Discussion section line 412-429 as follows “Previous studies suggest that psychological distress and perceived stress affect compliance negatively and non-compliance results in increased morbidity and mortality in kidney transplant recipients (6,7,11). Although an investigation about sleep quality and its related psychosocial variables among 438 renal transplant patients showed that the global PSQI scores were higher in participants with abnormal renal function compared with participants with normal renal function (35), there was no correlation between kidney function and PSS, PSQI, ISI, HAD-A and HAD-D scores in our study. Re-evaluation in short and long term follow-up of our study subjects during and after the Covid-19 pandemic will better reflect the association between sleep quality, stress levels and kidney transplant function.”

3. The importance of the study can at least be shown by citing previous research showing the importance of emotional well-being in kidney transplant recipients overall. Though a statement about this connection is made in abstract and introduction, there were no references provided. 

Response: We would like to thank reviewer for the kind comments related with our manuscript. We added a sentence into the Introduction section line 97-99 with provided reference as followed “Emotional well-being is important in kidney transplant recipients and improve the treatment compliance and decrease the probability of rejection (17).”

4. In addition, whether the stress levels are impacted by the Covid-19 pandemic was also shown, with moderate to weak evidence. 

Response: We would like to thank reviewer for the kind comments related with our manuscript. We agree with the reviewer and we mentioned this as a study limitation in the Discussion section line 439-441 as follows “Since the patients’ psychological status and sleep conditions before the out-break were not evaluated, it is difficult to infer a causal relationship between the variables of interest and the latter.” 

5. Finally, lack of correlation between PSS/PSQL/ISI/HAD readouts and kidney function from laboratory tests, shown in the results, does not add to the importance of the paper. 

Response: We would like to thank reviewer for the kind comments related with our manuscript. Transplanted patients tend to be more affected from stress disorders, insomnia, and poor sleep quality even before the Covid-19 pandemic. In fact, we investigated in this study, whether PSS/PSQL/ISI/HAD scores were affected more in those with poor kidney function. Unfortunately, we don’t know the impact of psychiatric and sleep disturbances of progression on kidney function related to Covid-19 pandemic in our kidney transplant recipients. We believe that it’s difficult to show correlation between PSS/PSQL/ISI/HAD score and kidney function at the moment. Re-evaluation after a certain time period of Covid-19 pandemic it would be better to show the connection between sleep quality, stress levels and kidney transplant function. 

6. Going into more detailed analysis, it would be very interesting to see any possible association between stress levels and hypertension, whose impact on kidney function is widely recognized. 

Response: We would like to thank reviewer for the kind comments related with our manuscript. We made more detailed analysis to see any possible association between stress levels, sleep disturbances and hypertension, whose impact on kidney function is widely recognized. We added the following paragraph to Results section line 290-293 as follows “When patients with hypertension (n:17) were compared to those without hypertension (n:89); there were no significant differences in PSS (24.3±6.4 vs 24.9±8.6, p: 0.784), PSQI (6.6±4.0 vs 6.0±3.8, p: 0.547), ISI (7.3±5.0 vs 6.5±5.7, p: 0.597), HAD-A (8.0±4.9 vs 7.0±4.1, p: 0.500) and HAD-D (7.8±3.2 vs 6.2±3.6, p: 0.114) scores between the two groups.” And we added following paragraph into the Discussion section line 430-437 “To address the impact of hypertension due to increased stress on kidney function we analysed the possible association between stress levels, sleep disturbances and hypertension. However, we failed to show any significant association between stress levels, sleep disturbances and hypertension. Xie et al. showed that the global PSQI scores were higher in the hypertension group compared with those in the non-hypertension group (35). The reason for the lack of association in our study may be due to the scarcity of hypertension among our subjects. Future studies may be needed to show the association after Covid-19 pandemic is over.”

7. And to make the paper more useful, it would help to identify what exactly about the pandemic (whether it is lack of human interaction, fear of socializing, loss of job, loss in the family) makes kidney transplant recipients prone to psychological disorders.

Response: We would like to thank reviewer for the kind comments related with our manuscript. Unfortunately, we did not ask any questions to identify what exactly about the pandemic makes kidney transplant recipients prone to psychological disorders. However, when we made a literature search we found some possible factors leading to psychiatric disorders in susceptible individuals such as kidney transplant recipients. We added a sentence into the Introduction section line 72-74 as follows “In the pandemic period, restrictions in social life, social isolation, quarantine, boredom, inadequate information, and virus-related fears can lead to various psychiatric disorders in susceptible individuals (3).”

8. My smaller notes would be about methods description and results presentation. It would be nice to know at which transplant center(s) the patients included in the study were followed-up. 

Response: We would like to thank reviewer for the kind comments related with our manuscript. We agree with reviewer. The transplant center is Marmara University. We added following sentence into the Methods and Materials section line 113-114 “The study population was selected among kidney transplant patients followed up at the Marmara University Hospital Transplantation clinic between 01.09.2020 and 01.12.2020.”

9. It would be also nice to see the breakdown of stress/sleep scores among patients grouped by creatinine/BUN levels. 

Response: We would like to thank reviewer for the kind comments related with our manuscript. We have grouped patients by GFR to see the breakdown of stress/sleep scores. We added following paragraph into the Results section line 279-283 as follows “When patients with less than 60 ml/min/1.73 m2 glomerular filtration rate (GFR) were compared to those with GFR>60 ml/min/1.73 m2; there were no significant differences in PSS (24.8±8.2 vs 24.5±9.4, p: 0.877), PSQI (6.2±3.5 vs 5.7±4.0, p: 0.547), ISI (7.3±5.4 vs 5.7±5.7, p: 0.168), HAD-A (7.4±4.6 vs 6.5±4.1, p: 0.357) and HAD-D (6.4±3.7 vs 6.7±3.6, p: 0.682) scores between the two groups.”

10. Most of tables should better be re-designed to focus on differences that are significant, rather than showing everything. 

Response: We would like to thank reviewer for the kind comments related with our manuscript. We completely agree with the reviewer and accordingly we have re-designed tables. We deleted Table 2 and re-ordered the statements of tables so Table 4 is named as Table 2. We changed Table 5 as Figure 1 and so Table 6 is named as Table 4. 

11. Table 3 can be presented such that patients are grouped by GFR and/or by measured stress levels. 

Response: We would like to thank reviewer for the kind comments related with our manuscript. We completely agree with the reviewer and accordingly we changed the Table 3, which is stated in the Results section, as patients grouped by measured stress levels high or low.

12. Some statements (such as “psychiatric disorders may lead to noncompliance” and “Outbreaks of infectious diseases and current Covid-19 may trigger…” in introduction) need to be provided with references. 

Response: We would like to thank reviewer for the kind comments related with our manuscript. We provided mentioned statements with references 11 and 16.

Reviewer2:

Is a very interesting paper looking at sleep disturbances and psychological issues in renal transplant recipients during the early phases of Covid 19

The authors study is subset of renal transplant recipients early in 2020.

This was early in the COVID-19 pandemic.

1. It is unclear what the baseline rate of sleep disturbances and concerns would be in this set of patients. For instance, "post holiday blues", other concerns with the renal transplant as well as family. 

Response: We would like to thank reviewer for the kind comments related with our manuscript. Transplanted patients tend to be more affected from stress disorders, insomnia, and poor sleep quality even before the Covid-19 pandemic. Unfortunately, we don’t know baseline rate of sleep disturbances and concerns of our kidney transplant recipients. We mentioned this as a study limitation in the Discussion section line 439-441 as followed “Since the patients’ psychological status and sleep conditions before the out-break were not evaluated, it is difficult to infer a causal relationship between the variables of interest and the latter.” And also after a literature search to identify what exactly about the pandemic makes kidney transplant recipients prone to psychological disorders we added a sentence into the Introduction section line 72-74 as follows “In the pandemic period, restrictions in social life, social isolation, quarantine, boredom, inadequate information, and virus-related fears can lead to various psychiatric disorders in susceptible individuals (3).” and We added a paragraph to show association the psychological impact of the pandemic and health conditions such as graft loss into the introduction section Line 75-81 as follows “Studies of previous quarantines for SARS, influenza A and Ebola revealed high rates of post-traumatic stress and depression up to 4 to 5 times higher in quarantined subjects 4. It has been shown before that transplant recipients are susceptible to anxiety, depression, and post-traumatic stress disorder 5. Some studies report that post-transplant depression and anxiety increases morbidities and mortality risk of the patients, with poorer medical adherence and/or pathophysiological abnormalities which contribute to poor health outcomes 6,7. We completely agree with the reviewer that it’s important to show "post holiday blues" and other concerns with kidney transplant recipients and as well as their family. We believe future studies may reveal the impact of pandemic on physiological and sleep disorders and its association between morbidity and mortality in kidney transplant recipients.

Journal Requirements

Response: We would like to thank editor for the kind comments related with our manuscript. We have checked our manuscript for PLOS ONE's style requirements.

2. Thank you for your ethics statement: "The investigation conforms with the principles outlined in the Declaration of Helsinki. The local ethics committee approved the study, and all participants gave written informed consent (Protocol number: 09.2020.991)."

Response: We would like to thank editor for the kind comments related with our manuscript. We have edited our current ethics statement as follows “The study design was approved by the institutional review board of Marmara University School of Medicine Ethic Committee and all participants gave written informed consent. (Protocol number: 09.2020.991).”

---

## [Editor Report · Decision Letter 1]

22 Feb 2021

The association between perceived stress with sleep quality, insomnia, anxiety and depression in kidney transplant recipients during Covid-19 pandemic

PONE-D-20-40967R1

Dear Dr. Atas,

We’re pleased to inform you that your manuscript has been judged scientifically suitable for publication and will be formally accepted for publication once it meets all outstanding technical requirements.

Kind regards,

Stanislaw Stepkowski

Academic Editor

PLOS ONE

Additional Editor Comments (optional):

None
---

## [Editor Report · Acceptance letter]

24 Feb 2021

PONE-D-20-40967R1 

The Association Between Perceived Stress with Sleep Quality, Insomnia, Anxiety and Depression in Kidney Transplant Recipients During Covid-19 Pandemic 

Dear Dr. Barutcu Atas:

I'm pleased to inform you that your manuscript has been deemed suitable for publication in PLOS ONE. Congratulations! Your manuscript is now with our production department. 

Kind regards, 

on behalf of

Dr. Stanislaw Stepkowski 

Academic Editor

PLOS ONE